# Regulation by cyclic di-GMP attenuates dynamics and enhances robustness of bimodal curli gene activation in *Escherichia coli*

**Olga Lamprecht**[¤ʘ]**, Maryia Ratnikava**[ʘ]**, Paulina Jacek, Eugen Kaganovitch, Nina Buettner, Kirstin Fritz, Ina Biazruchka**[iD]**, Robin Köhler, Julian Pietsch**[iD]**, Victor Sourjik**[iD]*

Max Planck Institute for Terrestrial Microbiology and Center for Synthetic Microbiology (SYNMIKRO), Marburg, Germany

ʘ These authors contributed equally to this work.
¤ Current address: Blood Transfusion Service Zurich, Swiss Red Cross (SRC), Schlieren, Switzerland
\* victor.sourjik@synmikro.mpi-marburg.mpg.de

**Data Availability Statement:** All data underlying our findings are included within the paper and its Supporting information files.

## Abstract

Curli amyloid fibers are a major constituent of the extracellular biofilm matrix formed by bacteria of the Enterobacteriaceae family. Within *Escherichia coli* biofilms, curli gene expression is limited to a subpopulation of bacteria, leading to heterogeneity of extracellular matrix synthesis. Here we show that bimodal activation of curli gene expression also occurs in well-mixed planktonic cultures of *E. coli*, resulting in all-or-none stochastic differentiation into distinct subpopulations of curli-positive and curli-negative cells at the entry into the stationary phase of growth. Stochastic curli activation in individual *E. coli* cells could further be observed during continuous growth in a conditioned medium in a microfluidic device, which further revealed that the curli-positive state is only metastable. In agreement with previous reports, regulation of curli gene expression by the second messenger c-di-GMP via two pairs of diguanylate cyclase and phosphodiesterase enzymes, DgcE/PdeH and DgcM/PdeR, modulates the fraction of curli-positive cells. Unexpectedly, removal of this regulatory network does not abolish the bimodality of curli gene expression, although it affects dynamics of activation and increases heterogeneity of expression levels among individual cells. Moreover, the fraction of curli-positive cells within an *E. coli* population shows stronger dependence on growth conditions in the absence of regulation by DgcE/PdeH and DgcM/PdeR pairs. We thus conclude that, while not required for the emergence of bimodal curli gene expression in *E. coli*, this c-di-GMP regulatory network attenuates the frequency and dynamics of gene activation and increases its robustness to cellular heterogeneity and environmental variation.

## Author summary

Formation of biofilms results in differentiation of bacterial matrix-embedded communities into several subpopulations that differ in their gene expression and physiology. Bacterial biofilms are typically associated with surfaces, but the underlying differentiation

**Funding:** This work was supported by the Max Planck Society (including salaries to OL, MR, PJ, EK, NB, KF, IB, JP, and VS) and by the Deutsche Forschungsgemeinschaft grant MU 4469/2-1 (including salary to RK). The funders had no role in study design, data collection and analysis, decision to publish, or preparation of the manuscript.

**Competing interests:** The authors have declared that no competing interests exist.

processes can also occur in planktonic cultures without or prior to surface attachment. Expression of biofilm genes is controlled by a multitude of regulatory inputs, with the second messenger cyclic di-GMP (c-di-GMP) being a common signal associated with the biofilm formation. Here we investigated the role of c-di-GMP regulation in activation of curli genes that encode a major protein constituent of the extracellular matrix of *Escherichia coli*, in planktonic and biofilm cultures and during growth in microfluidic channels. We observed that expression of curli genes is bimodal under all of these conditions, with bacteria differentiating into two distinct subpopulations of curli-positive and curli-negative cells even in absence of any external spatial cues. While regulation by c-di-GMP is not required for curli gene activation in a subpopulation of cells, it attenuates the frequency and dynamics of stochastic gene induction and reduces variability between single cells and different growth conditions. Thus, c-di-GMP signaling controls both probability and robustness of differentiation in a bacterial population.

## Introduction

Curli amyloid fibers are the key component of the extracellular matrix produced during biofilm formation by *Escherichia coli*, *Salmonella enterica*, and other Enterobacteriaceae [1–9]. In *E. coli* and *S. enterica* serovar Typhimurium, curli genes are organized in two divergently transcribed *csgBAC* and *csgDEFG* operons that share a common intergenic regulatory region [10]. Expression of these operons is under regulation by the stationary phase sigma factor $\sigma^S$ (RpoS), and it thus becomes activated during the entry into the stationary phase of growth [4, 11–14]. This activation is achieved by the $\sigma^S$-dependent induction of the transcriptional regulator CsgD, which then controls the expression of the *csgBAC* operon that encodes the major curli subunit CsgA along with the curli nucleator CsgB and the chaperone CsgC [7, 8, 15]. In turn, *csgD* expression in *E. coli* and *S.* Typhimurium is either directly or indirectly regulated by multiple cellular factors that mediate responses to diverse environmental changes, including both global and specific transcriptional regulators, small regulatory RNAs and second messengers (reviewed in [16–19]).

One of the key regulators of *csgD* is the transcription factor MlrA [13, 14, 20, 21]. The activity of MlrA depends on cellular levels of bacterial second messenger bis-(3'-5')-cyclic dimeric guanosine monophosphate (c-di-GMP), and in *E. coli* this control is known to be mediated by a pair of the interacting diguanylate cyclase (DGC) and phosphodiesterase (PDE) enzymes, DgcM and PdeR, that form a ternary complex with MlrA [12, 14, 22]. MlrA is kept inactive by binding PdeR, and this interaction is relieved when the latter becomes active as a PDE thus acting as the trigger enzyme [22, 23]. This inhibition is counteracted both by DgcM, which locally produces c-di-GMP to engage PdeR, as well as by the global pool of c-di-GMP. Besides its enzymatic activity, DgcM might also activate MlrA through direct protein interaction. Another DGC/PDH pair, DgcE and PdeH, provides global regulatory input into the local DgcM-PdeR-MlrA regulation [12, 24]. At least under typical conditions used to study *E. coli* biofilms, this regulatory network appears to be the only c-di-GMP-dependent input that controls *csgD* expression [24].

Previous studies of *E. coli* macrocolony biofilms formed on agar plates showed that curli expression occurs in the upper layer of the colony, but even in this layer its expression remained heterogeneous [25–27], indicating an interplay between global regulation of curli gene expression by microenvironmental gradients within biofilms and its inherent stochasticity. Differentiation of *E. coli* into distinct subpopulations of cells either expressing or not

expressing curli was also observed in submerged biofilms formed in liquid cultures, whereby curli expression was associated with cellular aggregation [28]. Furthermore, bi- or multimodality of *csgD* reporter activity was reported in the early stationary phase among planktonic cells in *S.* Typhimurium [29, 30] and *E. coli* [27]. Given established c-di-GMP-dependent regulation of CsgD activity, it was proposed that bistable curli expression originates from a toggle switch created by mutual inhibition between DgcM and PdeR, which could act as a bistable switch on *csgD* gene expression [27, 31].

In this study we demonstrate that differentiation of *E. coli csgBAC* operon expression, leading to formation of distinct subpopulations of curli-positive and -negative cells, occurs not only in submerged and macrocolony biofilms but also stochastically in a well-stirred planktonic culture and thus in the absence of any environmental gradients. Similar stochastic and reversible differentiation could further be observed among cells growing in conditioned medium in the microfluidic device. Although the c-di-GMP regulatory network consisting of the DgcE/PdeH and DgcM/PdeR pairs modulates the fraction of curli-positive cells in the population and the dynamics and uniformity of curli gene activation, it is not required to establish the bimodality of expression.

## Materials and methods

### Bacterial strains and plasmids

All strains and plasmids used in this study are listed in S1 Table. A derivative of *E. coli* W3110 [26] that was engineered to encode a chromosomal transcriptional sfGFP reporter downstream of the *csgA* gene [28] (VS1146) was used here as the wild-type strain. Gene deletions were obtained with the help of P1 phage transduction using strains of the Keio collection [32] as donors, and the kanamycin resistance cassette was removed using FLP recombinase [33]. For gene expression, *dgcE* and *pdeH* genes were cloned into the pTrc99A vector [34].

### Growth conditions for planktonic cultures

Planktonic *E. coli* cultures were grown in tryptone broth (TB) medium (10 g tryptone, 5 g NaCl per liter, pH 7.0) or lysogeny broth (LB) medium without salt (10 g tryptone, 5 g yeast extract, per liter, pH 7.0), supplemented with antibiotics where necessary. Overnight cultures were inoculated from LB agar plates, grown at 30°C and diluted 1:100, unless indicated otherwise, in 5–10 ml of fresh TB and grown at 30°C at 200 rpm in 100 ml flasks in a rotary shaker until the indicated $OD_{600}$ or overnight (18–25 h; $OD_{600}$ ~ 1.3–1.8). Alternatively, cultures were grown in TB in 96-well plates in a plate reader, with 200 μl culture per well, using cycles of 10 mins orbital shaking and 20 mins without shaking, except data in Fig 1A were alternating ~10 mins cycles of orbital and linear shaking were used instead to improve mixing. Where indicated, bacterial cultures were supplemented with either 1 mM L-serine (after 6 h of growth) or 0.1–10 mg/l DL-serine hydroxamate at inoculation.

### Growth and quantification of submerged biofilms

Submerged biofilms were grown and quantified as described previously [28], with minor modifications. Overnight bacterial cultures grown in TB were diluted 1:100 in fresh TB medium and grown at 200 rpm and 30°C in a rotary shaker to $OD_{600}$ of 0.5. The cultures were then diluted in fresh TB medium to a final $OD_{600}$ of 0.05, and 300 μl was loaded into a 96-well plate (Corning Costar, flat bottom; Sigma-Aldrich, Germany) and incubated without shaking at 30°C for 46 h.

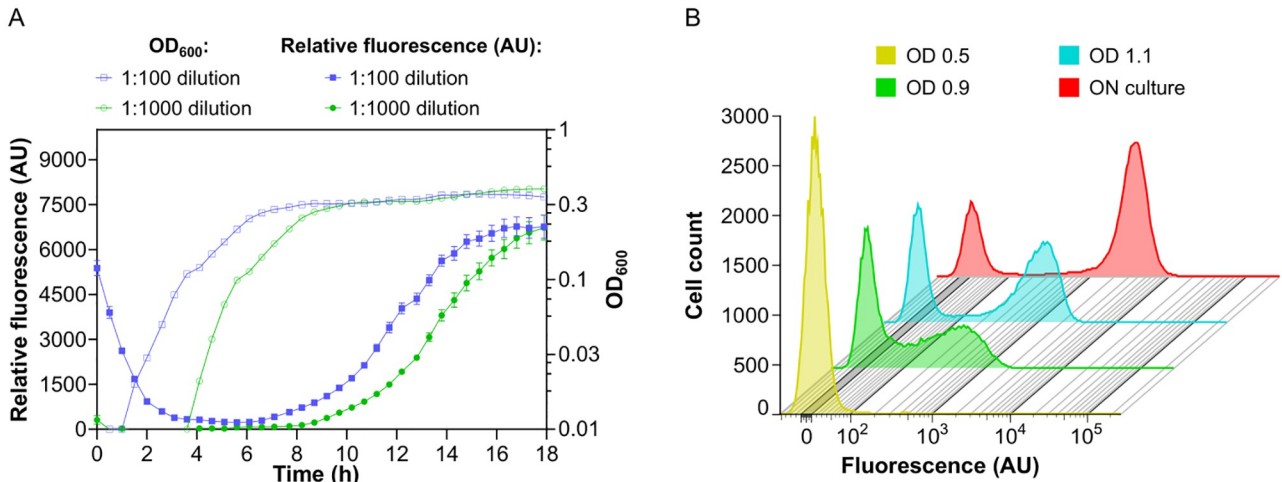

**Fig 1. Bimodal activation of curli gene expression in *E. coli* planktonic cultures.** *E. coli* cells carrying genomic transcriptional reporter of *csgBAC* operon were grown in liquid tryptone broth (TB) medium at 30°C under constant shaking. **(A)** Optical density ($OD_{600}$) and relative fluorescence (fluorescence/$OD_{600}$; AU, arbitrary units) of the culture during growth in a plate reader, starting from two different dilutions of the overnight culture. Error bars indicate standard error of the mean (SEM) of 10 technical replicates. **(B)** Distribution of single-cell fluorescence levels, measured by flow cytometry, of cultures grown from overnight culture diluted 1:1000 to indicated $OD_{600}$ or overnight (ON; 25 h) in flasks in an orbital shaker.

For quantification of biofilm formation, the non-attached cells were removed and the wells were washed once with phosphate-buffered saline (PBS; 8 g NaCl, 0.2 g KCl, 1.44 g $Na_2HPO_4$, 0.24 g $KH_2PO_4$). Attached cells were fixed for 20 min with 300 µl of 96% ethanol, allowed to dry for 40 min, and stained with 300 µl of 0.1% crystal violet (CV) solution for 15 min at room temperature. The wells were subsequently washed twice with 1x PBS, incubated with 300 µl of 96% ethanol for 35 min and the CV absorption was measured at $OD_{595}$ using INFINITE 200 PROplate reader (Tecan Group Ltd., Switzerland). Obtained CV values were normalized to the $OD_{600}$ values of the respective biofilm cultures.

## Macrocolony biofilm assay

Macrocolony biofilms were grown as described previously [26]. Briefly, 5 µl of the overnight liquid culture grown at 37°C in LB medium (10 g tryptone, 5 g NaCl, and 5 g yeast extract per liter) was spotted on salt-free LB agar plates supplemented with Congo red (40 µg/ml). Plates were incubated at 28°C for 8 days.

## Fluorescence measurements

Measurements of GFP expression in an INFINITE M1000 PRO plate reader (Tecan Group Ltd., Switzerland) were done using fluorescence excitation at 483 nm and emission at 535 nm. Relative fluorescence was calculated by normalizing to corresponding $OD_{600}$ values of the culture.

For fluorescence measurements using flow cytometry, aliquots of 40–300 µl of liquid bacterial cultures were mixed with 2 ml of tethering buffer (10 mM $KH_2PO_4$, 10 mM $K_2HPO_4$, 0.1 mM EDTA, 1 µM L-methionine, 10 mM lactic acid, pH 7.0). Macrocolonies were collected from the plate, resuspended in 10 ml of tethering buffer and then aliquots of 40 µl were mixed with 2 ml of fresh tethering buffer. All samples were vigorously vortexed and then immediately subjected to flow cytometric analysis using BD LSRFortessa Sorp cell analyzer (BD Biosciences, Germany) using 488-nm laser. In each experimental run, 50,000 individual cells were

analyzed. Absence of cell aggregation was confirmed by using forward scatter (FSC) and side scatter (SSC) parameters. Data were analyzed using FlowJo software version v10.7.1 (FlowJo LLC, Ashland, OR, US).

## Microfluidics

Conditioned medium was prepared by cultivating wild-type *E. coli* in TB in a rotary shaker at 30˚C for 20 h, after which the cell suspension was centrifuged at 4000 rpm for 10 min, medium was filter-sterilized and stored at 4˚C. Mother machine [35] microfluidics devices were fabricated using a two-layer soft lithography [36]. *E. coli* cells from the overnight culture in TB were loaded into the mother machine by manual infusion of the cell suspension through one of the two inlets using a 1-ml syringe. Cells were first allowed to grow at 30˚C for 4 h in fresh TB, then switched to the conditioned TB and cultivated for up to 26 h. Phase contrast and GFP fluorescence images were acquired using a Nikon Eclipse Ti-E inverted microscope with a time interval of 10 min. Cell segmentation was performed on phase contrast images using of a fully convolutional neural network based on the U-net architecture [37]. Details of microfluidics experiments and image analysis are described in S1, S2 and S3 Protocols.

## Results

### Bimodal curli gene expression is induced in planktonic culture

In order to characterize curli expression in planktonic culture of *E. coli*, we followed the induction of a chromosomal transcriptional reporter for the *csgBAC* operon, where the gene encoding stable green fluorescent protein (GFP) was cloned downstream of *csgA* with a strong ribosome binding site as part of the same polycistronic RNA [28]. In a previous study of submerged *E. coli* biofilms, this reporter showed bimodal expression both in the surface-attached biofilm and in the pellicle at the liquid-air interface, and its activity correlated with the recruitment of individual cells into multicellular aggregates [28]. When *E. coli* culture was grown at 30˚C in tryptone broth (TB) liquid medium with agitation, this reporter became induced during the transition to stationary phase (Fig 1A), which is consistent with previous reports [12, 14, 27]. The GFP reporter level remains high in the stationary phase, which also explains its initially high fluorescence in our plate reader experiments, where the culture was inoculated using stationary-phase cells. The observed induction of curli expression occurred at similar densities in the cultures with different initial inoculum size. The onset of induction apparently coincided with the reduction of the growth rate, which might occur due to depletion of amino acids in the medium and induction of the stringent response [38, 39], consistent with the proposed role of the stringent response in the regulatory cascade leading to curli gene expression [18, 23]. In agreement with that, curli expression was strongly reduced when *E. coli* cultures were grown in a concentrated TB medium (S1A Fig) or when TB medium was supplemented with serine (S2A Fig). Moreover, curli reporter induction was enhanced by the addition of serine hydroxamate (SHX), which is known to mimic amino acid starvation and induce the stringent response [40] (S2A Fig).

In order to investigate whether curli expression was uniform or heterogeneous within planktonic *E. coli* populations, we next measured curli reporter activity in individual cells using flow cytometry. The reporter was induced only in a fraction of cells, and this bimodality of curli expression became increasingly more pronounced at later stages of culture growth, reaching its maximum in the overnight culture (Fig 1B). Thus, the bimodal induction of curli gene expression is observed not only in biofilms but also in well-mixed planktonic cultures. Upon inspection of the flask culture by microscopy, no cell aggregation could be observed,

consistent with weakness of curli-mediated interactions [27] and suggesting that curli gene induction in a subpopulation of bacteria is not induced by the formation of suspended biofilm-like aggregates [41]. While curli activation was more pronounced in a cell culture growing in an orbital shaker in a flask (Fig 1B), bimodality was also observed for cultures grown in a plate reader (S1B and S2B Figs). Notably, stimulation of curli expression by SHX, or its suppression by additional nutrients, affected the fraction of positive cells rather than their expression levels (S1B and S2B Figs).

## Bimodality of curli gene expression does not require the c-di-GMP regulatory network composed of DgcE/PdeH and DgcM/PdeR pairs

Subsequently, we investigated dependence of bimodal activation of curli gene expression in planktonic culture on the known regulation of curli expression in *E. coli* by the global (DgcE and PdeH) and local (DgcM and PdeR) pairs of DGC/PDE enzymes [12, 14, 22] (Fig 2A). In agreement with this model of c-di-GMP-dependent regulation, activation of curli reporter was completely abolished in the absence of MlrA (Fig 2B), and it was strongly reduced by deletions of *dgcE* and *dgcM* and enhanced by deletions of *pdeH* and *pdeR* genes (Fig 2C). Notably, under these conditions a small but discernible fraction of curli-positive cells (~6%) could still be detected in cultures of both DGC gene deletion strains, and a small fraction of curli-negative cells (1–3%) was observed for both PDE gene deletion strains, indicating that deletions of these genes do not eliminate the bimodality of curli gene expression or its levels in curli-positive cells but rather change the probability of curli genes to become activated in individual cells.

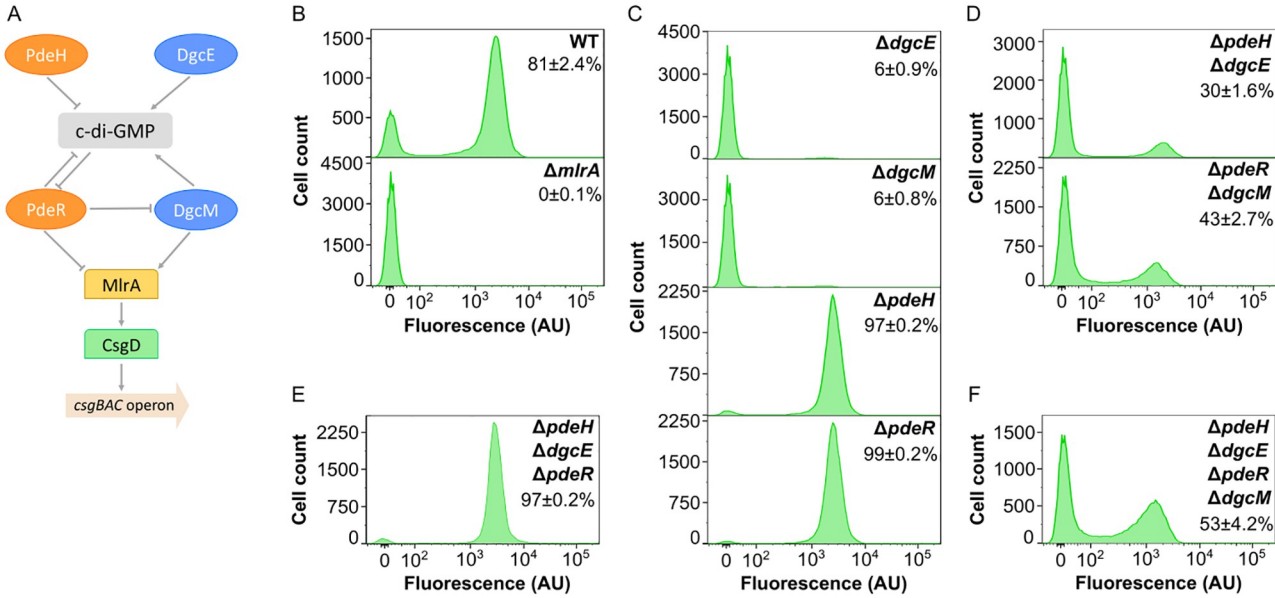

**Fig 2. Regulation of *csgBAC* operon expression by c-di-GMP. (A)** Current model of regulation of curli gene expression by c-di-GMP in *E. coli*, adapted from [27]. The regulation is mediated by two pairs of diguanylate cyclases (DGCs; blue) and phosphodiesterases (PDEs; orange). PdeH and DgcE control global levels of c-di-GMP, whereas PdeR and DgcM mediate local c-di-GMP-dependent regulation of curli gene expression by controlling activity of transcription factor MlrA, which activates another curli-specific transcription factor CsgD. Finally, CsgD controls expression of the *csgBAC* operon that encodes the major curli subunit CsgA. **(B-F)** Flow cytometry measurements of *csgBAC* expression in *E. coli* planktonic cultures grown in TB overnight in flasks in an orbital shaker, shown for the wild-type (WT) and Δ*mlrA* strain (B), and individual (C), double (D), triple (E) and quadruple (F) deletions of DGC or PDE enzymes, as indicated. Fraction of positive cells in the population (mean of three biological replicates ± SEM) is indicated for each strain. Note that the scale in the *y* axes is different for individual strains to improve readability.

Consistently, bimodality was also retained upon combined deletions of pairs of DGC/PDE genes. Removal of the entire global level of c-di-GMP regulation led to a bimodal pattern of curli reporter activation within the population of Δ*pdeH* Δ*dgcE* cells (Fig 2D) that was similar to that observed in the wild-type culture, despite a smaller fraction of curli-positive cells in the deletion strain. Even more surprisingly, the distribution of curli reporter expression remained bimodal upon removal of the local level of c-di-GMP regulation in Δ*pdeR* Δ*dgcM* strain, although the fraction of curli-positive cells was reduced in this background, too.

We observed that level of curli gene expression in Δ*pdeR* Δ*dgcM* strain was lower than in Δ*pdeR* strain, which confirms that DgcM can promote curli expression independently of PdeR [12, 14, 22] (Fig 2A). This conclusion is further supported by the comparison between the triple deletion strain Δ*pdeH* Δ*dgcE* Δ*pdeR* that expresses only DgcM (Fig 2E) and the quadruple deletion strain Δ*pdeH* Δ*dgcE* Δ*pdeR* Δ*dgcM* that lacks the entire regulatory network (Fig 2F), with much higher fraction of curli-positive cells observed in the former background.

Most importantly, the bimodality of curli expression was still observed in the quadruple deletion strain (Fig 2F). We thus conclude that, whereas the known c-di-GMP-dependent regulation of MlrA activity does clearly affect the fraction of curli-positive cells, it is not required to establish bimodality of curli gene expression in the cell population. The network has further only little impact on the level of curli reporter activity in individual positive cells (i.e., on position of the positive peak in the flow cytometry data), although reporter intensity in positive cells appeared to be slightly reduced in the strains lacking DgcM.

*Vibrio cholerae* transcriptional regulator VpsT, a close homologue of CsgD, has been shown to be directly regulated by binding to c-di-GMP [42]. Furthermore, in both S. Typhimurium and *E. coli*, curli gene expression might also be regulated by c-di-GMP independently of *csgD* transcription [43–45]. We thus aimed to examine whether *E. coli* curli gene expression was no longer sensitive to the global cellular level of c-di-GMP in the absence of the local PdeR/DgcM regulatory module. Indeed, whereas the overexpression of DgcE or of PdeH, which both affect the global pool of c-di-GMP in *E. coli*, had strong impacts on the fraction of curli-positive cells in the wild type, the quadruple mutant was insensitive to such overexpression (Fig 3). This result suggests that in this background, the expression of the *csgBAC* reporter is indeed no longer affected by global c-di-GMP levels. This further excludes regulation by other DGCs that contribute to the common c-di-GMP pool, although it does not rule out hypothetical regulation by a local DGC/PDE pair that is insensitive to global c-di-GMP levels.

## Curli gene activation shows higher growth-condition dependent variability in the absence of the c-di-GMP regulatory network

We next explored how the fraction of curli-positive cells in the population depends on the conditions of culture growth, with or without regulation by c-di-GMP. As noted above, when the wild-type *E. coli* culture was grown in multi-well plates, both the fraction of curli-positive cells and the level of reporter activity in positive cells were moderately reduced compared to incubation in a flask in an orbital shaker (Figs 2B, 4A and S1B). The reduction in the number of curli-positive cells under these growth conditions was even more pronounced for Δ*pdeR* Δ*dgcM* or Δ*pdeH* Δ*dgcE* Δ*pdeR* Δ*dgcM* strains (Figs 4A and S3), where only a small fraction of cells (12%) became positive, compared to 40–50% in the flask culture (Fig 2D and 2F). Interestingly, this difference was less for the Δ*pdeH* Δ*dgcE* strain (24% vs 30%). Deletions of individual DGC or PDE genes generally showed expected phenotypes, but fractions of curli-positive cells in Δ*pdeH* and Δ*pdeR* strains were again reduced compared with flask cultures (Fig 2C). Another notable difference between flask and multi-well plate cultures was that the low-fluorescence peak of the wild-type culture was not fully negative but apparently contained a large

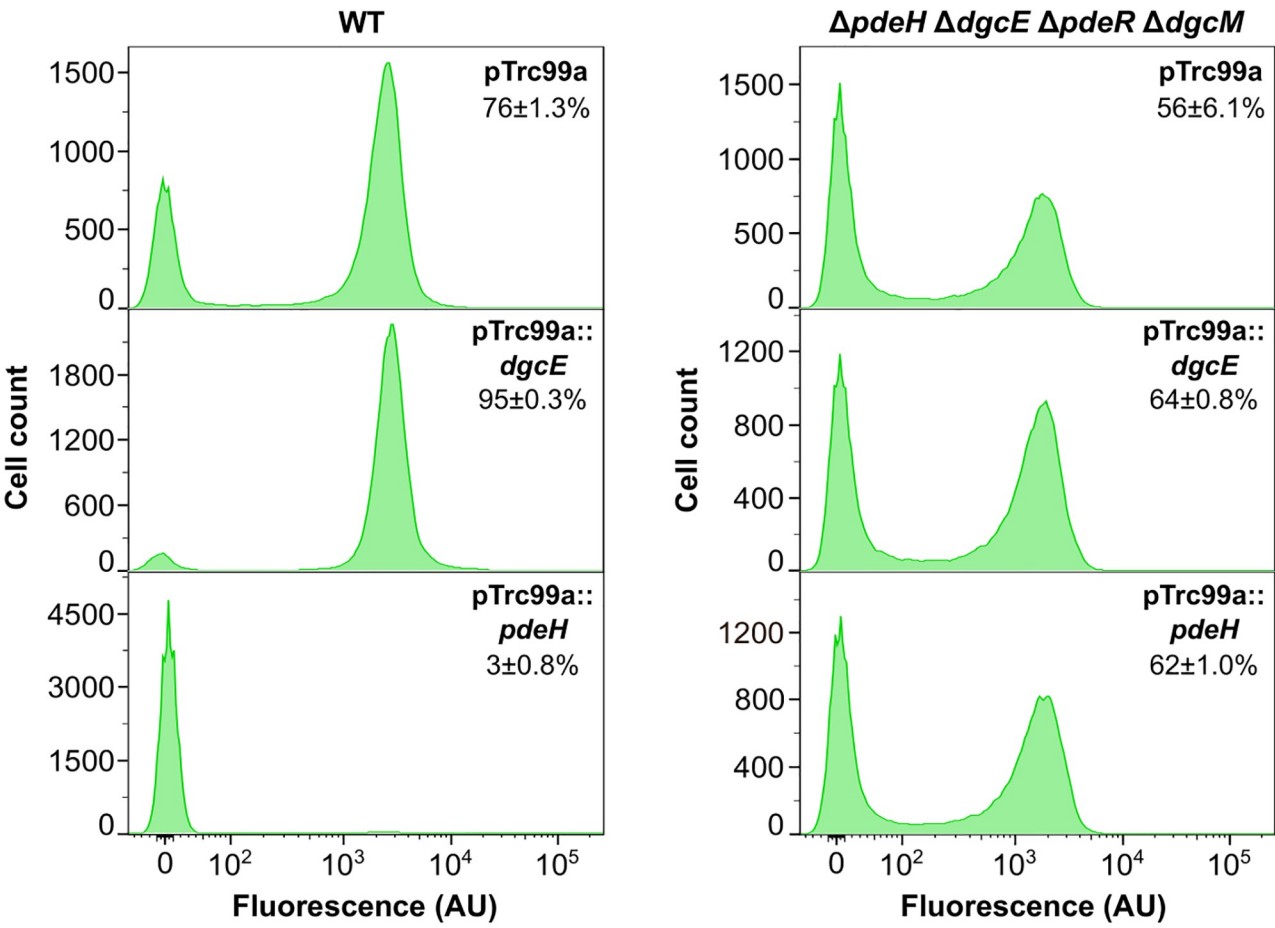

**Fig 3. Decoupling of curli gene expression from c-di-GMP regulation in the absence of PdeR/ DgcM regulatory module.** *E. coli* wild-type (WT) cells or cells lacking c-di-GMP regulatory enzymes (Δ*pdeH* Δ*dgcE* Δ*pdeR* Δ*dgcM*) were transformed with either empty pTrc99a plasmid (control) or with pTrc99a plasmid carrying *dgcE* (pTrc99a::*dgcE*) or *pdeH* (pTrc99a::*pdeH*) genes. Gene expression was induced with 1 μM IPTG. Bacteria were grown in TB overnight in flasks with shaking and cultures were subjected to the flow cytometry analysis. Fraction of positive cells in the population (mean of three biological replicates ± SEM) is indicated for each strain.

fraction of cells with incompletely activated curli reporter, which could also be seen in the Δ*pdeH* or Δ*pdeR* strains but not in the Δ*pdeH* Δ*dgcE* Δ*pdeR* Δ*dgcM*, Δ*pdeR* Δ*dgcM* or Δ*pdeH* Δ*dgcE* strains. Similar results were obtained even upon prolonged incubation in the plate reader (S4 Fig), confirming that the observed difference with the overnight flask culture was not because of the different growth stage but rather due to differences in growth conditions.

We further tested reporter activation under growth conditions that favor biofilm formation. During formation of static submerged biofilms in multi-well plates, where cultures are grown without shaking, the overall curli activation pattern in cell populations of individual or double and quadruple deletion strains (Fig 4B) was comparable to that in the flask culture grown in the orbital shaker (Fig 2B–2D and 2F), although effects of DGC and PDE gene deletions on the fraction of curli-positive cells were apparently reduced. Curli gene activation in individual mutant strains correlated well with the levels of submerged biofilm formation (S5 Fig), but the lack of regulation by c-di-GMP resulted in stronger reduction of the biofilm biomass than simply expected from its effect on curli gene expression, consistent with additional roles of c-di-GMP in biofilm formation.

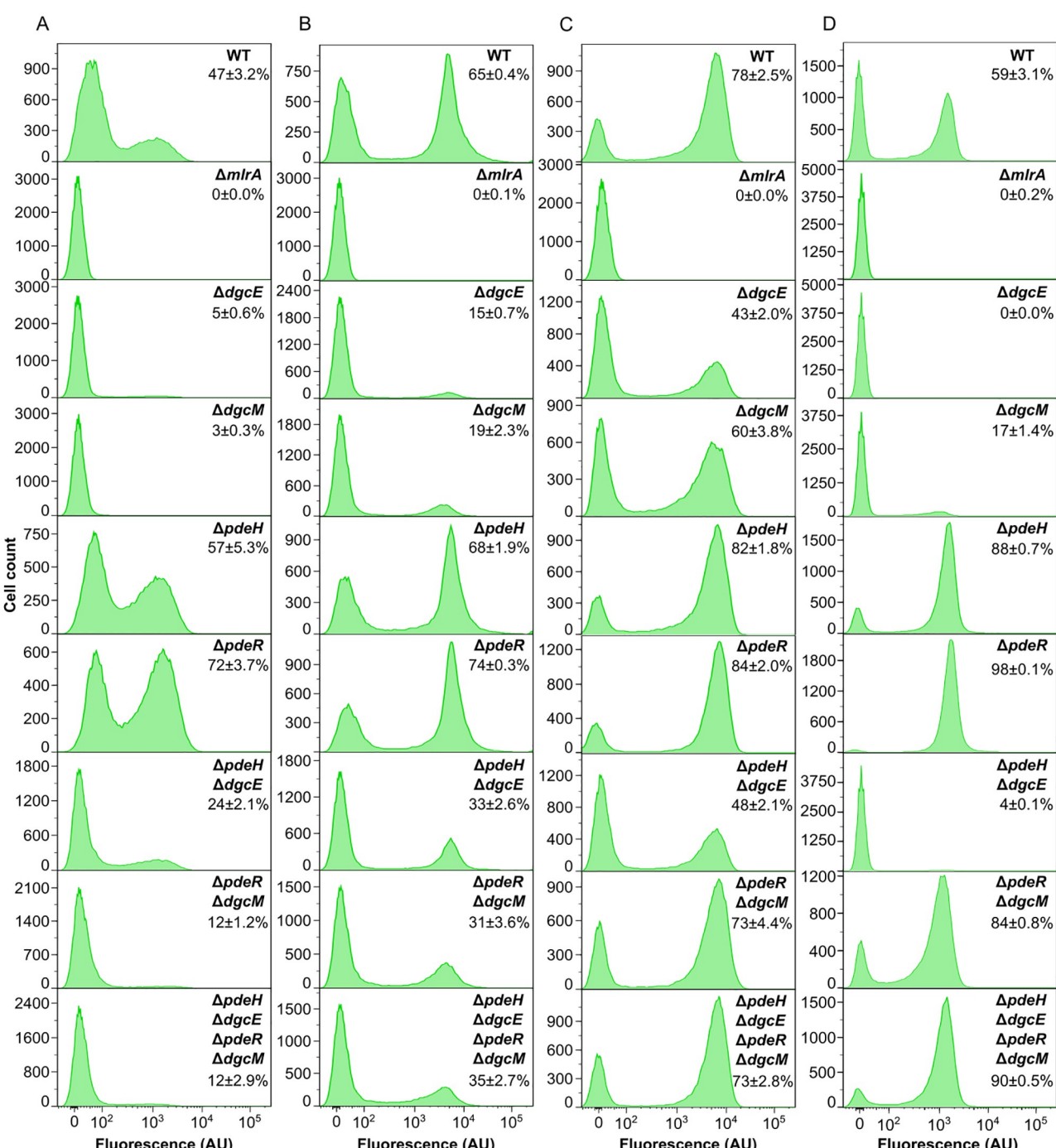

**Fig 4. Dependence of curli gene expression on c-di-GMP regulation under different growth conditions. (A-D)** Flow cytometry measurements of curli expression in the wild-type (WT) and indicated gene deletion strains either after 24 h of growth in liquid TB in a plate reader (A) or as submerged biofilms without agitation (B), as macrocolony biofilms on salt-free LB agar (C), or as overnight planktonic cultures in salt-free LB in flasks in an orbital shaker (D). Fraction of positive cells in the population (mean of three biological replicates ± SEM) is indicated for each strain.

All strains were also grown in the form of macrocolony biofilms on agar plates containing the salt-free LB medium, as done previously [26] (Fig 4C), as well as in the liquid salt-free LB medium as a control (Fig 4D). Expression of curli reporter in individual cells within the macrocolony was analyzed by flow cytometry immediately after colony resuspension (see Materials and methods), so that the level of stable GFP reporter should reflect cell-specific curli expression within the macrocolony. The overall dependency of reporter activation on the deletions of individual DGC and PDE genes was again qualitatively similar to other growth conditions (Fig 4C). However, the fraction of curli-positive cells in the macrocolony biofilms was less sensitive to deletions of individual *dgc* or *pdh* genes than in planktonic cultures, regardless whether the latter were grown in TB (Fig 2C) or in the salt-free LB (Fig 4D). This agrees with stronger Congo red staining of macrocolonies of Δ*dgcE* and Δ*dgcM* strains compared to the Δ*mlrA* negative control (S6 Fig). Thus, during macrocolony biofilm formation, regulation by the network composed of DgcE/PdeH and DgcM/PdeR pairs appears to be less important for the activation of curli gene expression, possibly because other regulators contribute to this activation in a highly structured environment at the level of *mlrA*, *csgDEFG* or *csgBAC* operons. Importantly, the curli-positive cell fraction in both macrocolony and planktonic cultures of Δ*pdeR* Δ*dgcM* and Δ*pdeH* Δ*dgcE* Δ*pdeR* Δ*dgcM* strains growing on the salt-free LB was much larger than under other growth conditions.

To summarize, under all of the tested growth conditions, regulation by the known c-di-GMP regulatory network was not required for bimodal curli activation, although it indeed affected the fraction of curli-positive cells. Nevertheless, the absence of this c-di-GMP control apparently makes the fraction of curli-positive cells in the population more sensitive to environmental conditions, since it exhibited much larger variation in the quadruple deletion compared to the wild-type strain among different growth conditions tested here.

## Dynamics and variability of stochastic curli gene activation in individual cells is controlled by the c-di-GMP regulatory network

In order to investigate the dynamics of curli gene activation, and the impact of c-di-GMP regulation, at the single-cell level, we utilized a "mother machine" device—a microfluidic chip where growth of individual bacterial cell lineages could be followed in a highly parallelized manner over multiple generations [35] (Fig 5 and S1 Protocol). Since our design of the mother machine allows rapid switching of the medium supplied to the cells (Fig 5D), we could mimic nutrient depletion in order to activate curli expression in continuously growing single cells. For that, wild-type, individual DGC or PDE, or quadruple deletion strains were first loaded into the mother machine from overnight cultures, allowed to grow in fresh TB medium for several generations, and then shifted to TB medium that was pre-conditioned by growing a batch culture (see Materials and methods and S2 Protocol). A fraction of curli-positive cells was observed at the beginning of these experiments, which was strain-specific and consistent with the expected fraction of positive cells in the overnight cultures of each respective strain. After resuming exponential growth in the fresh medium all cells turned off curli expression (Figs 6A, 6C and S7 and S1, S2 and S3 Movies and S3 Protocol). Following the shift to conditioned medium, cell growth rate was strongly reduced (Figs 6E and S8), and after several generations of slow growth, individual cells of all strains activated curli expression, while other cells remained in the curli-off state (Figs 6A, 6C, 6F and S7 and S1, S2 and S3 Movies). Importantly, although the absolute frequencies of curli-positive cells in individual mutants were somewhat different in the mother machine from those in the planktonic cultures grown in the flask (Fig 2), likely due to differences in growth conditions as observed already for different batch cultures and biofilms (Fig 4), the relative frequencies between strains were consistent,

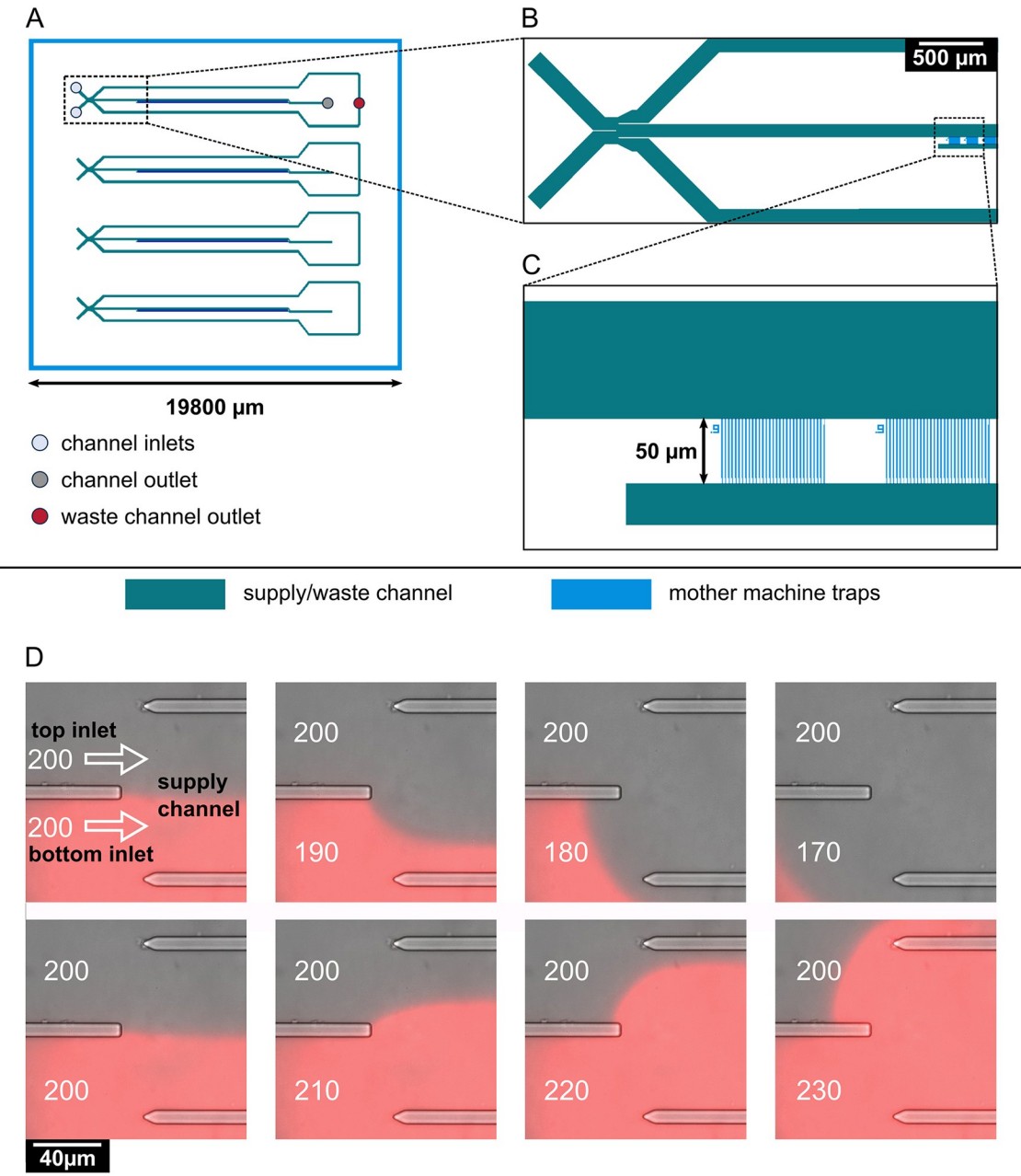

**Fig 5. Design and operation of the microfluidic mother machine device. (A)** Schematic overview of the chip layout, featuring four independent supply channels (green) for cell inoculation and media supply. **(B)** Detailed view of the area marked by a rectangle in (A), showing the switching junction and a part of the mother machine cultivation sites (blue). The junction is formed by two inlets, leading to one central supply channel. Control of pressure at each inlet allows for prioritization of one medium over the other through the supply channel, and, ultimately, the mother machine cultivation sites. Residual medium flows out through waste channels located to each side of the central supply channel. Medium flowing through the supply channel exits the chip through one outlet. **(C)** Detailed view of the area marked in **(B)** by a rectangle, showing the mother machine cultivation sites. Each of the four channels contains 57 mother machine cultivation sites, each of which contains 30 mother machine traps with widths of either 0.9, 1, or 1.1 µm. The mother machine traps feature a 0.3 µm wide constriction on the bottom, preventing the mother cell from exiting the trap while allowing perfusion of the medium. The supply channels (green) are 8 µm in depth, the mother machine traps (blue) are 0.8 µm in depth. **(D)** On-chip medium switching visualized by merged phase contrast and mCherry images of the channel junction. Media are supplied through separate inlets (top and bottom), which are separated in the center of the channel by a PDMS barrier. The direction of flow is indicated by white arrows. Water was supplied through the top inlet, while a 0.2 µM sulforhodamine B solution was supplied through the bottom inlet, visualizing the flow pattern in the junction. The pressure at the top inlet was kept constant at 200 mbar. Depending on the pressure set at the bottom inlet, it is possible to select which one of the two media flows into the central supply channel to the mother machine growth sites.

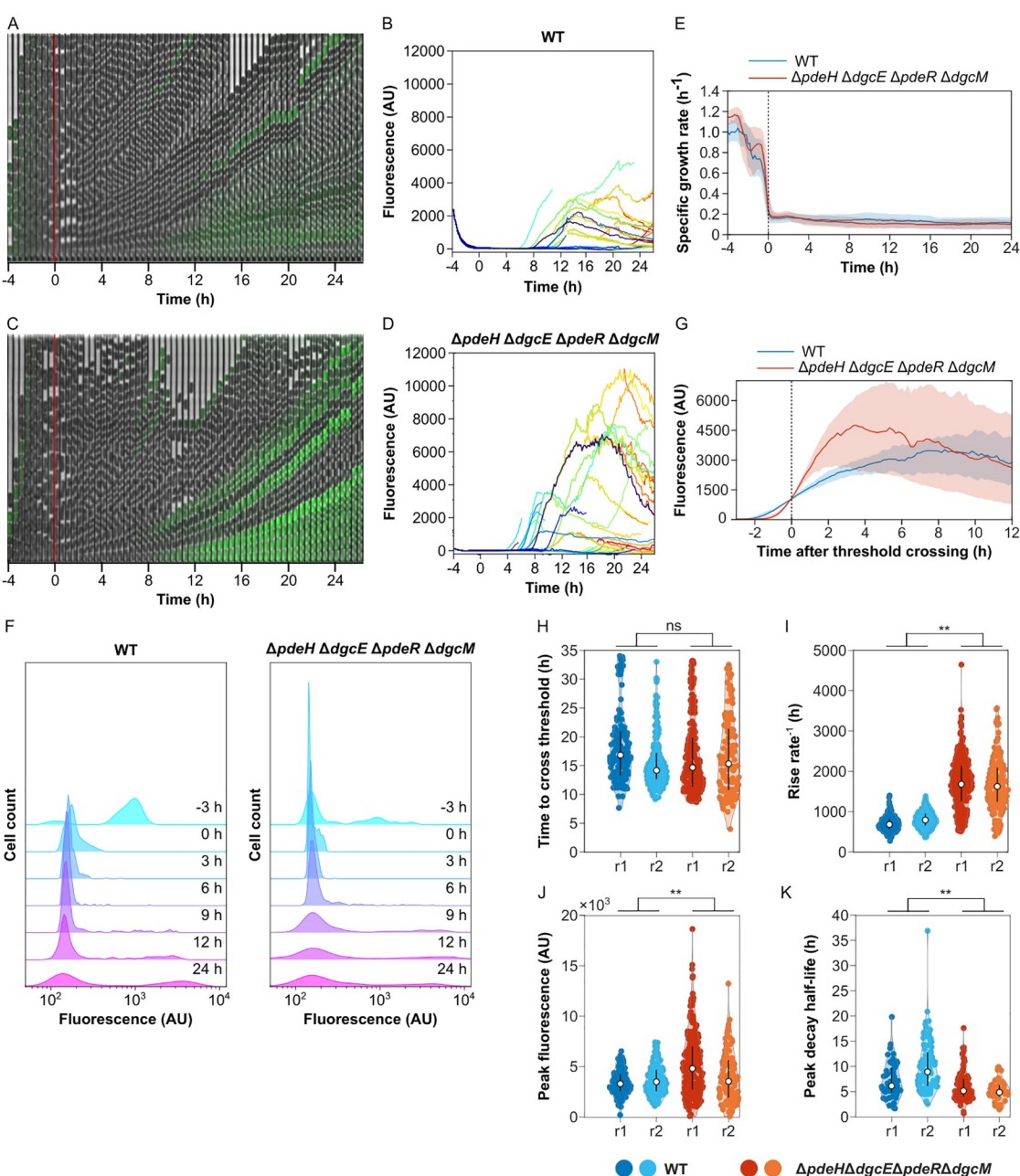

**Fig 6. Impact of c-di-GMP regulation on dynamics of curli gene induction in individual cells.** *E. coli* cells in a microfluidic device (mother machine) were shifted from a fresh to conditioned TB medium after 4 h of growth to induce curli expression. **(A-D)** Examples of image time series and single-cell fluorescence traces for the wild-type (WT) (A,B) and for the *ΔpdeH ΔdgcE ΔpdeR ΔdgcM* strain disabled in the c-di-GMP regulation (C,D), growing in a mother machine in one experiment. Expression of the curli reporter is indicated by the green overlay on the phase contrast image. **(E)** Median instantaneous growth rate—fold rate of change in length—of cells grown in microfluidics experiments as described in (A). The switch to conditioned medium is at time zero, as indicated. Shaded area is the interquartile range. The number of cells in the device varies with time, but is on average $n = 296$ for WT and $n = 522$ for the quadruple deletion strain. **(F)** Distributions of curli expression at different time points in the microfluidics experiment. Shown are kernel density estimates of curli expression in WT and in the quadruple deletion strain at selected time points. **(G)** Median curli expression profile for single-cell traces aligned by the time at which they exceed a threshold of $10^3$ fluorescence units. Shaded area is interquartile range; $n = 128$ for WT and 230 for the quadruple deletion strain. **(H-K)** Distributions of curli induction parameters for the wild-type and for the c-di-GMP-regulation disabled strain, in two independent experiments, r1 (same experiment as in A-G) and r2. Shown are histograms of the times at which a threshold of $10^3$ fluorescence units were crossed **(H)**, the maximum rates of increase in fluorescence **(I)**, the fluorescence amplitudes at the first peak **(J)** and the half-time of fluorescence decay after reaching the peak **(K)** for cell traces from both microfluidics experiments. $^{**}p < 10^{-2}$ in an unpaired two-sample *t*-test assuming normal distributions with unknown and unequal variances. ns, non-significant.

confirming that the overall control of curli gene expression by the c-di-GMP regulatory network is similar for growth in conditioned medium in the mother machine.

Although the overall frequency of curli reporter activation was comparable between the wild-type and Δ*pdeH* Δ*dgcE* Δ*pdeR* Δ*dgcM* cells lacking the c-di-GMP regulatory network, we further compared the dynamics of single-cell curli activation in both backgrounds. First, we observed that, in both strains, curli reporter activation was transient—and after several generations in the curli-on state individual cells turned curli expression off again during continuous growth in the conditioned medium (Figs 6B, 6D, S8, and S9). In some cases, the initial transient pulse of activation was even followed by a second activation event. Although such transient curli reporter activation was observed in both backgrounds, with a similar delay after exposure to conditioned medium (Fig 6H), its dynamics showed several important differences between the strains. Most clearly, the rate of curli activation in individual cells was higher in the absence of the c-di-GMP regulatory network (Figs 6G, 6I, S10A and S10B), which led to stronger but more transient increase in the curli reporter fluorescence (Figs 6J, 6K and S10B). Additionally, the level of reporter induction showed greater intercellular heterogeneity in the Δ*pdeH* Δ*dgcE* Δ*pdeR* Δ*dgcM* strain (Figs 6D, 6J and S9). Thus, the control of curli expression by c-di-GMP reduces the rate but also the cell-to-cell variability of curli gene induction within population.

## Discussion

Expression of the curli biofilm matrix genes is known to be heterogeneous or even bistable in communities of *E. coli* [25–28] and *S.* Typhimurium [29, 30], which might have important functional consequences for the biomechanics of bacterial biofilms [27] and for stress resistance and virulence of bacterial populations [29, 30]. In the well-structured macrocolony biofilms, differentiation into subpopulations with different levels of curli matrix production is largely deterministic and driven by gradients of nutrients and oxygen [18], but bimodality of curli expression might also emerge stochastically, in a well-mixed population or between cells within the same layer of a macrocolony [27, 29, 30]. How this bimodality originates within the extremely complex regulatory network of curli genes [17, 19, 23, 27] remains a matter of debate. Generally, it is well established that the bimodality, and subsequently bistability, of gene expression in bacteria can be achieved by stochastic gene activation combined with an ultrasensitive response of the regulatory network, for example due to a positive transcriptional feedback loop or positive cooperativity [46–48]. Although earlier work proposed that bistable expression of curli might be a consequence of positive transcriptional feedback in *csgD* regulation [30, 49], the most recently suggested model attributed bistability to the properties of the potentially ultrasensitive c-di-GMP regulatory switch formed by DgcM, PdeR and MlrA [27, 31]. Furthermore, although this bimodal pattern of curli gene expression has typically been referred to as bistable, temporal dynamics and stability of its induction in individual cells has not been directly investigated.

Here, we studied the expression of the major curli *csgBAC* operon in well-stirred planktonic *E. coli* cultures, as well as in submerged and macrocolony biofilms, and during growth in a microfluidics device. Consistent with previous studies [12, 14, 27], the induction of curli gene expression in growing *E. coli* cell populations was observed during entry into the stationary phase of growth. Curli activation was apparently dependent on depletion of nutrients, most likely amino acids, since it could be suppressed by increasing the levels of nutrients or, more specifically, by addition of serine. Conversely, it could be enhanced by the SHX-mediated stimulation of the stringent response, which mimics amino acid starvation. We further observed that activation of the *csgBAC* operon exhibited strongly pronounced bimodality under all tested conditions, resulting in two distinct, curli-positive and -negative,

subpopulations of cells even in a well-stirred planktonic culture. Stochastic bimodality of curli reporter activation was confirmed in a microfluidic device, where, following shift to conditioned medium, only a fraction of the cell population turned on curli gene expression.

Such differentiation is apparently consistent with previous reports of bimodal *csgD* expression in the stationary phase of *S.* Typhimurium [29, 30] and *E. coli* [27] culture growth. However, in contrast to the previous interpretation of such a *csgD* expression pattern as bistability, we observed that activation of curli gene expression during continuous cell growth in the microfluidic device was only transient and unstable. Moreover, our data indicate that the bimodality of the *csgBAC* expression is unlikely to be (fully) explained by the bimodal expression of *csgD*. Firstly, compared to the bimodality of the *csgBAC* reporter, the previously reported bimodality of *csgD* expression in *E. coli* [27] was much less pronounced and became apparent only in later stages of planktonic culture growth. Moreover, in that study, all wild-type cells showed activation of the *csgD* reporter, differing only in the level of this activation, which itself cannot explain the on/off bimodality of the *csgBAC* reporter activation.

Secondly, and most importantly, whereas the bimodality of *csgD* expression was observed to be dependent on regulation by the DgcE/PdeH/DgcM/PdeR network [27], this was clearly not the case for *csgBAC* reporter expression. Under all tested conditions, including planktonic cultures, submerged and macrocolony biofilms, and growth in the microfluidic device, the differentiation into distinct subpopulations of positive and negative cells in our experiments occurred in the absence of this entire c-di-GMP regulatory network. Since removal of the DgcM/PdeR module further made curli gene expression insensitive to global levels of c-di-GMP, contributions of other *E. coli* DGCs and PDEs to curli regulation in this deletion background are unlikely, even if these enzymes might in principle affect curli gene expression in wild-type *E. coli* cells via the common c-di-GMP pool. We therefore conclude that c-di-GMP control might be dispensable for activation or bimodality of curli gene expression in *E. coli*, although we could not rule out regulation of curli gene expression by a hypothetical local DGC/PDE module that is insensitive to global c-di-GMP levels.

Nevertheless, c-di-GMP control clearly plays an important role during the establishment of bimodality. Consistent with previous studies, we observed that the DGC and PDE proteins determine the fraction of curli-positive cells [22, 23, 27], although they have no or only little effect on the average level of *csgBAC* expression in individual cells. Our results further suggest that the DgcE/PdeH/DgcM/PdeR regulatory network might make curli gene expression in *E. coli* populations more robust, helping to ensure that the fraction of curli-positive cells is less variable under different culture growth conditions. This might be related to the observed effect of this network on the temporal expression dynamics in a continuously growing culture, with faster but more heterogeneous activation of curli gene expression in the absence of the c-di-GMP control, although whether and how there two different kinds of variability are connected remains to be investigated. It is further possible that the increased robustness provided by c-di-GMP regulation is due to the network-induced bimodality of *csgD* expression discussed above [27], which might provide an additional stabilizing feedback.

How might the observed pulsatile activation of the curli-positive state originate at the single-cell level and lead to the differentiation into distinct subpopulations in *E. coli* culture? Pulsing in expression was proposed to be common to many gene regulatory circuits [50–52]. It was recently described in *E. coli* for the upstream regulator of curli, $\sigma^S$ [53], as well as for the flagellar regulon [54, 55] that is anti-regulated with curli [28]. However, in neither of these cases did pulsing lead to bimodality of expression, and their relation to the observed pulses in curli expression thus remains to be seen. As mentioned above, in addition to stochastic pulsing, the emergence of bimodality requires an ultrasensitive network response, which could be achieved through one or several positive transcriptional feedbacks. Given the dependence of

*csgBAC* operon expression on MlrA and CsgD, such feedback is likely to involve these two transcription factors, and possibly other regulators of *mlrA*, *csgDEFG* or *csgBAC* operons.

Regardless of the origin of pulsatile expression, the timing and duration of curli activation pulses can explain how such transient activation leads to differentiation into two distinct sub-populations in the planktonic culture. Since the curli expression is only turned on during transition to the stationary phase, individual cells stochastically activate curli genes just before the culture growth ceases, thus making subsequent reversion to the curli-negative state impossible. Although curli gene regulation in biofilms might be more complex, spatial stratification of gene expression in the macrocolony biofilms was proposed to resemble the temporal regulation of expression in planktonic cultures [18, 26]. Thus, also in this case, curli activation would happen in a cell layer just before it ceases to grow, fixing the differentiation between curli-positive and curli-negative cells within this colony layer.

## Supporting information

**S1 Table. *E. coli* strains and plasmids used in this study.**
(PDF)

**S1 Fig. Dependence of curli gene expression on nutrient levels.** Wild-type E. coli cultures were grown as in Fig 1A (except shaking conditions) but with different indicated concentrations of TB. (A) Bacterial growth and activity of transcriptional curli reporter. Error bars indicate SEM of 6 technical replicates. (B) Distribution of single-cell fluorescence levels after 24 h of growth in a plate reader measured by flow cytometry. Note that the scale in the y axes is different for individual conditions to improve readability.
(PDF)

**S2 Fig. Stimulation of curli gene expression by stringent response.** Wild-type E. coli cultures were grown as in Fig 1A (except shaking conditions) but with addition of either indicated concentrations of serine hydroxamate (SHX) at inoculation point or 1 mM serine after 6 h of growth. (A) Bacterial growth and activity of transcriptional curli reporter. Error bars indicate SEM of 6 technical replicates. (B) Distribution of single-cell fluorescence levels after 24 h of growth in a plate reader measured by flow cytometry.
(PDF)

**S3 Fig. Relative fluorescence of curli reporter in planktonic culture grown in TB medium in a plate reader.** Error bars indicate SEM of 5 technical replicates. E. coli planktonic cultures of the wild-type (WT), ΔmlrA strain, and individual, double and quadruple deletions of DGC or PDE enzymes were grown in TB in a plate reader as in Fig 1A (except shaking conditions).
(PDF)

**S4 Fig. Curli gene expression upon prolonged cultivation in a plate reader.** Wild-type (WT) and ΔpdeH ΔdgcE ΔpdeR ΔdgcM strains were grown in TB in a plate reader as in S3 Fig, but during 36 h. (A) Induction of transcriptional curli reporter. Error bars indicate SEM of 10 technical replicates. (B) Distribution of single-cell fluorescence levels in populations of both strains after 36 h of growth measured by flow cytometry.
(PDF)

**S5 Fig. Curli gene expression in submerged biofilm cultures.** Biofilm formation by indicated strains grown in multi-well plates in TB medium, quantified using crystal violet (CV) staining. Error bars indicate SEM of 3 independent replicates.
(PDF)

**S6 Fig. Curli expression in macrocolony biofilms.** Images of macrocolonies of indicated strains after 8 days of growth on salt-free LB agar plates.
(PDF)

**S7 Fig. Distributions of curli expression at different time points in the microfluidics experiment.** Shown are kernel density estimates of curli expression in the wild-type WT, individual and quadruple deletions of DGC or PDE enzymes at selected time points. Cells in stationary phase were loaded into mother machine chip supplied with fresh medium, and then switched to the conditioned media at the "0 h" time point. Note that the scale in the y axes is different for individual conditions to improve readability.
(PDF)

**S8 Fig. Growth rates and fraction of curli expressing cells over time.** Stationary phase cells were introduced into mother machine devices, supplied with fresh medium and then switched to conditioned medium after 4 h of growth, as in Fig 6. (A) Median instantaneous growth rates for the wild-type and for the ΔpdeH ΔdgcE ΔpdeR ΔdgcM strain disabled in c-di-GMP regulation. Growth rate drops rapidly and cells switch on curli expression after a switch to conditioned medium. Shaded area is interquartile range. (B) Fraction of cells with fluorescence exceeding 1000 units. (C) Number of detected cells. Two biological replicates (r1 and r2) were performed for each strain; data for the r1 replicate are also shown in Fig 6. Note that in the r2 experiment for the quadruple deletion strain, cells were only imaged after medium switching.
(PDF)

**S9 Fig. Single-cell traces of cell fluorescence for all cells for the wild-type and for the c-di-GMP-regulation disabled strain.** Data are from the same biological replicates (r1 and r2) as in S8 Fig.
(PDF)

**S10 Fig. The rate and variability of curli induction for the wild-type and for the c-di-GMP-regulation disabled strain.** (A) Median curli expression, (B) production rate, and (C) number of cells for traces from both microfluidics experiments (r1 and r2) aligned by the time at which they exceeded a threshold of 103 fluorescence units. Shaded area is interquartile range. Compared to the WT, the rate of curli induction is faster but traces show more variability in a mutant without the global or local c-di-GMP regulatory modules.
(PDF)

**S1 Protocol. Design and fabrication of the mother machine.**
(PDF)

**S2 Protocol. Growth experiments in the microfluidics device.**
(PDF)

**S3 Protocol. Analysis of the microfluidics data.**
(PDF)

**S1 Data. Numerical data that underly graphs in the manuscript.**
(XLSX)

**S1 Movie. Activation of curli genes in DGC gene knockouts.** Wild-type, Δ*dgcE* or Δ*dgcM E. coli* cells, as indicated, carrying GFP reporter of *csgBAC* expression were shifted in the mother

machine from a fresh to conditioned TB medium after 4 h of growth to induce curli gene expression.
(AVI)

**S2 Movie. Activation of curli genes in PDH gene knockouts.** Wild-type, Δ*pdeH* or Δ*pdeR E. coli* cells, as indicated, carrying GFP reporter of *csgBAC* expression were shifted in the mother machine from a fresh to conditioned TB medium after 4 h of growth to induce curli gene expression.
(AVI)

**S3 Movie. Activation of curli genes in quadruple gene knockout.** Wild-type or Δ*pdeH* Δ*dgcE* Δ*pdeR* Δ*dgcM E. coli* cells, as indicated, carrying GFP reporter of *csgBAC* expression were shifted in the mother machine from a fresh to conditioned TB medium after 4 h of growth to induce curli gene expression.
(AVI)

## Acknowledgments

We thank Sarah Hoch, Silvia González Sierra and Gabriele Malengo for technical help and advice.

## Author Contributions

**Conceptualization:** Olga Lamprecht, Eugen Kaganovitch, Victor Sourjik.

**Data curation:** Julian Pietsch.

**Formal analysis:** Maryia Ratnikava, Julian Pietsch.

**Funding acquisition:** Victor Sourjik.

**Investigation:** Olga Lamprecht, Maryia Ratnikava, Paulina Jacek, Eugen Kaganovitch, Nina Buettner, Kirstin Fritz, Ina Biazruchka, Julian Pietsch.

**Methodology:** Olga Lamprecht, Paulina Jacek, Eugen Kaganovitch, Kirstin Fritz, Robin Köhler, Julian Pietsch.

**Project administration:** Victor Sourjik.

**Resources:** Eugen Kaganovitch.

**Supervision:** Victor Sourjik.

**Visualization:** Maryia Ratnikava, Eugen Kaganovitch, Ina Biazruchka, Julian Pietsch, Victor Sourjik.

**Writing – original draft:** Olga Lamprecht, Maryia Ratnikava, Paulina Jacek, Eugen Kaganovitch, Julian Pietsch, Victor Sourjik.

**Writing – review & editing:** Olga Lamprecht, Maryia Ratnikava, Julian Pietsch, Victor Sourjik.

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
