## [Decision Letter · Decision Letter 0]

26 Oct 2022

Dear Dr Sourjik,

Thank you very much for submitting your Research Article entitled 'Regulation by cyclic-di-GMP attenuates dynamics and enhances robustness of bimodal curli gene activation in Escherichia coli' to PLOS Genetics.

The manuscript was fully evaluated at the editorial level and by two of the independent peer reviewers who evaluated the prior submission. The reviewers appreciated the attention to an important problem, but raised some substantial concerns about the current manuscript. Based on the reviews, we will not be able to accept this version of the manuscript, but we would be willing to review a much-revised version. We cannot, of course, promise publication at that time.

If you decide to revise the manuscript for further consideration at PLOS Genetics, please aim to resubmit within the next 60 days, unless it will take extra time to address the concerns of the reviewers, in which case we would appreciate an expected resubmission date by email to plosgenetics@plos.org.

We are sorry that we cannot be more positive about your manuscript at this stage. Please do not hesitate to contact us if you have any concerns or questions.

Yours sincerely,

Gregory P. Copenhaver

Editor-in-Chief

PLOS Genetics

Reviewer's Responses to Questions

**Comments to the Authors:**

Reviewer #1: This is the revised version of a manuscript previously submitted to PloS Genetics.

While the description of the work has significantly improved, I still have the following comments (not considering that some of the controls like assessment of catalytic mutants of cyclic di-GMP turnover proteins have not been performed in any case).

First, to state that cyclic di-GMP does not substantially affect bimodality (bistability) is not necessarily justified as this work shows that deletion of individual PDEs in two instances shifts >99% (?) of cells to expression. Where is the threshold for definition of a bimodal/bistable expression (at a specific growth condition) and how do genes behave that are not bimodally expressed? Is there any reference for this?

The control experiment with overexpression of a diguanylate cyclase to demonstrate ‘blindness’ to cyclic di-GMP involves the same DGC as deleted in the regulatory pathway. Another DGC which shows potent DGC activity such as AdrA or a constitutively activated response regulator-DGC is necessary to make an initial conclusion. It has been demonstrated in screens (examples can be found in Kulasakara et al, PNAS, 2006) that despite of high cyclic di-GMP production, cells can be ‘blind’ to the observed physiological changes and vice versa.

Despite the rigorous conclusions drawn and, admittedly sophisticated experimental methodology, the initial theoretical design is rather sloppy as the MlrA, PdeH/DgcE and PdeR/DgcM regulatory network has been shown for csgD expression while expression of the csgBA operon is monitored. To the knowledge of this reviewer it has not even been formally demonstrated whether mlrA is involved in transcription of the csgBA operon. My suspicion would be that it is not.

Other comments:

References end at reference 47, but more are cited in the text.

Curli only expression leads to cell aggregation, but these interactions can be easily disrupted.

l. 186: do you mean: …formation of suspended biofilm-like aggregates is not correlated with curli gene induction…

l. 201: which percentage?

l.223: …in the cell population.

l.313: In the movie S2, in a pdeH deletion mutant less csgA expression seem to be observed. Why?

Reviewer #2: In this revised manuscript, Lamprecht et al. analyze the bimodal activation of curli expression using a transcriptional reporter for the csgBAC operon and flow cytometry. A main finding is described: bimodality of curli expression does not require the upstream modules formed by the DGC/PDE pairs DgcE/PdeH and DgcM/PdeR, although this network attenuates the frequency and dynamics of gene activation and increases its robustness to changing environmental conditions. I thank the authors for their extensive revision of the work, modification of the text and additional controls. Most of my previous concerns were addressed adequately. However, there are still minor points that remain to be addressed.

1. Abstract: line 30: the study can not rule out the involvement of other DGCs and thus c-di-GMP synthesized by them, in regulation of csgBAC expression. I suggest to change the sentence to “.. .shows stronger dependence on growth conditions in the absence of DgcE/PdeH and DgcM/PdeR regulation.”

2. Line 193: For more clarity, I suggest to change the title of this section to: Bimodality of curli gene expression does not require the c-di-GMP regulatory network comprised of the DGC/PDE pairs DgcE/PdeH and DgcM/PdeR.

3. Line 218: Change culri to curli

4. Line 224: Change: of the level…. To “on the level”.

5. All along the study, results indicate that in the absence of the regulatory network composed of DgcE/PdeH and DgcM/PdeR, MlrA is free to activate csgD transcription, and thus csgBAC expression, although is activity is lower than in the wild type strain because of the absence of DgcM. Since MlrA does not bind c-di-GMP, global changes in c-di-GMP do not impact curli expression or bimodality in the absence of DgcM/PdeR (Fig 3). Therefore, bimodality in the absence of this network seems to be caused by bimodality in mlrA expression. I understand that analyzing the biomodality of mlrA expression is out of the scope of this manuscript, but this issue should at least be discussed along the article.

6. Also, although curli expression may be regulated by c-di-GMP independently of csgD transcription, the study shows that a mlrA mutant is completely negative for curli expression and that bimodality disappears. This also should be discussed.

7. Line 283: Change “regulation by c-di-GMP” to “regulation by the regulatory network composed of DgcE/PdeH and DgcM/PdeR”

8. Line 284: again, results could be due to a reduced bimodality of mlrA expression under microcolony biofilm forming conditions.

**Have all data underlying the figures and results presented in the manuscript been provided?**

Reviewer #1: Yes

Reviewer #2: Yes

PLOS authors have the option to publish the peer review history of their article (what does this mean?). If published, this will include your full peer review and any attached files.

Reviewer #1: No

Reviewer #2: No

---

## [Decision Letter · Decision Letter 1]

18 Apr 2023

Dear Dr Sourjik,

We are pleased to inform you that your manuscript entitled "Regulation by cyclic-di-GMP attenuates dynamics and enhances robustness of bimodal curli gene activation in Escherichia coli" has been editorially accepted for publication in PLOS Genetics. Congratulations!

Yours sincerely,

Gregory P. Copenhaver

Editor-in-Chief

PLOS Genetics

Comments from the reviewers (if applicable):

Reviewer's Responses to Questions

**Comments to the Authors:**

Reviewer #1: Besides some instances where the deletion of the two synthase/phosphodiesterase pairs is still equalized with the entire cyclic di-GMP network , I think that the manuscript is fine.

Reviewer #2: The authors have substantially improved the manuscript. All my previous concerns were addressed adequately and thus, for this revised manuscript, I do not have any further concerns.

**Have all data underlying the figures and results presented in the manuscript been provided?**

Reviewer #1: Yes

Reviewer #2: Yes

PLOS authors have the option to publish the peer review history of their article (what does this mean?). If published, this will include your full peer review and any attached files.

Reviewer #1: No

Reviewer #2: No

**Data Deposition**

http://datadryad.org/submit?journalID=pgenetics&manu=PGENETICS-D-22-01106R1

**Press Queries**

---

## [Editor Report · Acceptance letter]

9 May 2023

PGENETICS-D-22-01106R1 

Regulation by cyclic di-GMP attenuates dynamics and enhances robustness of bimodal curli gene activation in *Escherichia coli*

Dear Dr Sourjik, 

We are pleased to inform you that your manuscript entitled "Regulation by cyclic di-GMP attenuates dynamics and enhances robustness of bimodal curli gene activation in *Escherichia coli*" has been formally accepted for publication in PLOS Genetics! Your manuscript is now with our production department and you will be notified of the publication date in due course.

With kind regards,

Anita Estes

PLOS Genetics

On behalf of:
